# Effects of parental migration on early childhood development of left-behind children in Bangladesh: Evidence from a nationally representative survey

**Shimlin Jahan Khanam** *, **Md Nuruzzaman Khan**

Jatiya Kabi Kazi Nazrul Islam University, Mymensingh, Bangladesh

* shimlinjahan2208@gmail.com

## Abstract

### Background

In Bangladesh, as in other low- and middle-income countries, parents frequently migrate to other areas, often for employment opportunities, leaving their children behind with the hope that their earnings will contribute to securing a better future for them. However, the absence of parents due to migration can have negative implications for the well-being of these left-behind children. Despite the existence of studies investigating this phenomenon, the evidence thus far has produced inconclusive findings, with no specific data available from Bangladesh. Therefore, the objective of this study was to examine the effects of parental migration on the early childhood development of left-behind children in Bangladesh.

### Methods

The present study utilized data from the Bangladesh Multiple Indicator Cluster Survey (MICS) conducted in 2019. A sample of 8,833 children aged 3–4 years was included in the analysis. The Early Childhood Development Index (ECDI) and its individual domains served as the outcome variables of interest. The primary explanatory variables considered in the analysis were father migration, mother migration, migration of both parents, and migration of either parent. To assess the association between the outcomes and explanatory variables, multilevel logistic regression analysis was employed, controlling for relevant covariates.

### Results

Approximately 29% of all children in the study were not developmentally on track, as measured by the ECDI. When examining the individual domains of the ECDI, only 9% of the total children demonstrated developmental progress in the learning domain. Regarding the association between parental migration and ECDI outcomes, we observed a 26% decrease in the likelihood of overall ECDI among children with a migrated father (OR: 0.74, 95% CI: 0.54–0.93). This decrease became even more pronounced, reaching 37% (OR: 0.63, 95%

**Data Availability Statement:** The data supporting the findings of this study are accessible through UNICEF but are not publicly available. Researchers interested in accessing the dataset can do so by

submitting a research proposal to UNICEF, similar to the process we followed to obtain the dataset for this study. The dataset can be accessed at https://mics.unicef.org/surveys. Interested researchers can apply to access the datasets at https://mics.unicef.org/visitors/sign-up.

**Funding:** The author(s) received no specific funding for this work.

**Competing interests:** The authors have declared that no competing interests exist.

CI: 0.48–0.97), among children with both parents migrated, compared to children with neither parent migrated.

## Conclusion

The findings of this study indicate that parental migration, especially when both parents are involved, has a substantial negative impact on the likelihood of achieving favourable ECDI outcomes for children in Bangladesh. To ensure the optimal development of children with migrating parents, it is crucial to strengthen early childhood development education programs and implement robust social safety nets. These efforts should specifically target the unique needs and challenges faced by children with migrated parents, providing them with the necessary support and resources for their holistic development.

## Background

An estimated 244 million children are left behind (defined as individuals under 18 years of age whose parents have relocated to another place for a period exceeding six months) globally, around 163 millions of them live in low- and middle-income countries (LMICs) [1, 2]. The Asia-Pacific region, including Bangladesh with an estimated 6 million left-behind children, has the highest number of such children, totalling an estimated 118 million, followed by sub-Saharan Africa (33 million) and Latin America (12 million) [2]. The primary reason for this migration, which results in children being left behind, is that parents frequently embark on migration journeys in pursuit of enhanced economic prospects [3, 4]. Left behind children confront a multitude of challenges stemming from their separation from parents, social isolation, and economic disadvantage [1, 5]. While migration can yield economic benefits for families remaining in the home country, the aforementioned difficulties can profoundly impact the cognitive, socio-emotional, and physical development of these children [4].

Early childhood development (ECD) is a critical period of growth and learning that sets the foundation for future success and well-being [6, 7]. The acquisition of key skills such as language, motor, cognitive, and socio-emotional abilities during early childhood is crucial for children to reach their full potential [7, 8]. However, left-behind children may be at a disadvantage in terms of ECD due to a range of factors such as inadequate care, poor nutrition, lack of stimulation, and social isolation [4, 8]. Yet, research exploring the impact of parental migration on ECD outcomes among left-behind children in LMICs, and particularly in the context of Bangladesh, remains limited. Previous studies have primarily focused on the impact of parental migration on children's health, education, and psychological well-being, yielding mixed findings [1, 9–13]. Moreover, the few studies that have specifically explored the consequences of parental migration on ECD outcomes have encountered several limitations, such as small sample sizes and variations in the factors considered as confounders [3, 10, 13, 14]. Additionally, it is essential to consider the unique cultural and social context of each country, including Bangladesh, as findings from one country may not necessarily apply to others [4]. Hence, there is a need to investigate the specific effects of parental migration on ECD outcomes for left-behind children in Bangladesh. This study aims to fill this research gap by examining the influence of parental migration on the ECD of left-behind children in Bangladesh, utilizing nationally representative survey data.

## Methods

### Study setting and data

We conducted an analysis of secondary data derived from the 2019 Multiple Indicator Cluster Survey (MICS) in Bangladesh [15]. This comprehensive survey took place from January to June 2019. It is worth noting that the MICS is a household survey developed by UNICEF, conducted in various countries, with the aim of providing statistically rigorous data on a wide range of indicators concerning the situation of children and women. In Bangladesh, 2019 MICS survey was carried out by the Bangladesh Bureau of Statistics in collaboration with the UNICEF and other partners. Covering all 64 districts of Bangladesh, the survey utilized a two-stage stratified cluster sampling design. In the first stage, 1,300 primary sampling units (PSUs) were selected. These PSUs were defined as either rural or urban areas with an approximate population of 200 to 300 households. The second stage involved the systematic random sampling of 28 households from each PSU, resulting in a total sample size of 36,400 households. Within these households, 24,686 eligible children were identified, and data were collected from 23,099 of these children with a response rate of 93.60%.

### Analytical sample

We have included children aged 3–4 years old from whom the ECDI-related questions were asked, following the guidelines provided by UNICEF [15, 16]. Out of the total 23,099 children included in the main survey, there were 8,833 children aged 3–4 years old, with 4,513 children aged 3 years and 4,320 children aged 4 years. Therefore, they were included in the analysis.

### Outcome variable

The study focused on the Early Childhood Development Index (ECDI), a composite measure encompassing a ten-item module across four domains: literacy-numeracy (3 items), physical (2 items), social-emotional (3 items), and learning (2 items) [16, 17]. Children were deemed on track in the literacy-numeracy domain if they achieved milestones like letter identification, reading simple words, and number recognition. For the physical domain, on-track status depended on fine motor skills and limited illness hindrance. Positive interaction with peers and non-aggressive behavior were indicators in the social-emotional domain. In the learning domain, following directions and independent task completion were considered on track. Domain indexes were calculated based on specific criteria recommended by UNICEF, with at least two positive responses needed in literacy-numeracy and social-emotional domains, two positive replies in the physical domain, and at least one positive reply in the learning domain for a positive classification [16, 17]. The ECDI was then derived as the percentage of children meeting developmental milestones in at least three of these domains, distinguishing between those who were developmentally on track and those who were not. This comprehensive index is widely utilized by organizations, including UNICEF, to monitor progress towards early childhood development goals [16, 17].

### Explanatory variable

The primary explanatory variable examined was the parental migration status. The relevant data pertaining to parental migration status were collected during the survey by posing specific questions to eligible respondents, typically the caretakers of the children. The questions included inquiries such as "Where does the natural father of (name of the selected children) reside?" and "Where does the natural mother of (name of the selected children) reside?" If an affirmative response was recorded, additional follow-up questions were asked to determine the

duration (in months) that the natural father or mother had lived away from home. Based on these responses, variables were generated to indicate father migration and mother migration, with a positive response indicating the absence of the respective parent for at least 6 months. Additionally, based on these two other migration variables were created: one representing any type of parental migration (either father or mother migrated, versus none), and the other representing both parents' migration (both father and mother migrated, versus none). These classifications were based on previous studies conducted in Bangladesh [4].

## Covariates

Numerous covariates were carefully selected to adjust potential confounding effects on the association between parental migration and the early childhood development of left-behind children in Bangladesh. The selection of these covariates was based on prior research findings [4, 6, 8–10, 14]. The covariates taken into account consist of the child's age, sex, birth order, and health status; household socioeconomic status (SES); maternal education and employment status; as well as the presence of other adult caregivers in the household.

## Statistical analysis

Descriptive statistics were utilized to provide a summary of the study population's characteristics, including age, sex, socioeconomic status, and other pertinent variables. To examine the relationship between parental migration status (father migration, mother migration, both parents' migration) and ECDI of left-behind children, multilevel logistic regression analysis was conducted, controlling for covariates. The decision to employ a multilevel logistic regression model was based on the hierarchical structure of the MICS data, where individuals were nested within households, and households were nested within clusters. This nesting structure resulted in a linkage between individual responses, rendering simple logistic regression inadequate for producing accurate results. In contrast, the multilevel logistic regression model was deemed the recommended approach for generating more robust outcomes [18]. The findings were reported as odds ratios (OR) with 95% confidence intervals (95% CI). A p-value of less than 0.05 was considered statistically significant. All analyses were performed using the statistical package STATA version 15.1.

**Ethics approval and consent to participate.** The survey data analyzed in this study are part of the Multiple Indicator Cluster Surveys conducted by UNICEF worldwide. The survey underwent ethical review by both UNICEF and the National Research Ethics Committee of the Bangladesh Medical Research Council. Written informed consent was obtained from all participants and/or their parents/guardians where applicable. All necessary patient/participant consents have been obtained, and the relevant institutional forms have been archived. No separate ethical approval was required to conduct this study. We obtained permission to access this survey and conduct this research.

## Results

### Demographic characteristics of the respondents

The demographic characteristics of the respondents are presented in Table 1. The study included an almost equal number of children aged 3 year and 4 year, as well as an equal distribution of males and females. Approximately 18% of the total children participated in early childhood education. It was found that around 45% of the children's mothers had attained a secondary level of education, while 4% of the children's mothers reported functional difficulties. Roughly one-fourth of the children were from households classified as the poorest.

**Table 1. Demographic characteristics of the respondents, N = 8,833.**

| Demographics of children | Percentage (95% CI) |
|---|---|
| **Age (in years)** | |
| 3 | 50.3 (49.1–51.6) |
| 4 | 49.7 (48.4–50.9) |
| **Sex** | |
| Male | 49.8 (49.0–50.7) |
| Female | 50.2 (49.4–51.0) |
| **Attendance in early childhood education** | |
| No | 82.0 (80.9–83.0) |
| Yes | 18.0 (17.0–19.1) |
| **Children mother's education** | |
| Pre-primary education | 16.7 (15.6–17.9) |
| Primary education | 26.7 (25.5–27.9) |
| Secondary education | 45.2 (43.9–46.6) |
| Higher education | 11.4 (10.6–12.2) |
| **Mother's functional difficulties** | |
| Has functional difficulties | 3.6 (1.5–4.5) |
| Has no functional difficulties | 96.4 (95.9–96.8) |
| **Wealth Index** | |
| Poorest | 25.0 (23.7–26.5) |
| Second | 20.8 (19.8–22.0) |
| Middle | 18.2 (17.3–19.3) |
| Fourth | 18.0 (16.9–19.1) |
| Richest | 17.9 (16.7–19.1) |
| **Place of residence** | |
| Urban | 20.1 (19.0–21.3) |
| Rural | 79.9 (78.8–81.0) |
| **Place of region** | |
| Barishal | 5.6 (5.2–6.0) |
| Chattogram | 23.5 (22.3–24.6) |
| Dhaka | 22.4 (21.3–23.6) |
| Khulna | 9.4 (8.9–10.0) |
| Mymensingh | 8.1 (7.3–8.9) |
| Rajshahi | 11.4 (10.7–12.1) |
| Rangpur | 10.3 (9.6–10.9) |
| Sylhet | 9.4 (8.5–10.4) |

Moreover, approximately four-fifths of the children lived in rural areas, and Chattogram division was reported as the region of residence by a similar proportion of the children.

## Distribution of early childhood development and parental migration situation in Bangladesh

Table 2 presents the distribution of domains of ECDI and parental migration. The findings revealed that 71% of the total children were found to be developmentally on track in terms of ECDI. In the independent domains of ECDI, approximately 73% and 78% of the total children demonstrated developmental progress in the literacy-numeracy and physical domains, respectively. Around 34% of the total children exhibited developmental progress in the socio-

**Table 2. Descriptive statistics exposure and outcome variables.**

| Children who are not developmentally track for indicated domains | % (95% CI) |
|---|---|
| **Children who are developmentally on track for indicated domains** | |
| Literacy-numeracy | 73.5 (72.4–74.7) |
| Physical | 78.4 (77.3–79.5) |
| Socio-emotional | 33.7 (32.5–34.9) |
| Learning | 9.1 (8.3–9.9) |
| Early child development index | 71.0 (69.8–72.1) |
| **Migration status of parents** | |
| **Father migration** | |
| Not migrated | 88.2 (87.3–89.0) |
| Migrated abroad | 8.5 (7.8–9.5) |
| Migrated within country | 3.3 (2.9–3.7) |
| **Mother migration** | |
| Not migrated | 97.7 (97.4–98.2) |
| Migrated abroad | 0.3 (0.1–0.04) |
| Migrated within country | 2.0 (1.7–2.3) |
| **Migration of both parents** | |
| No | 98.1 (97.8–98.4) |
| Yes | 1.9 (1.6–2.2) |
| **Any type of migration** | |
| Yes | 12.3 (11.5–13.2) |
| No | 87.7 (11.5–13.2) |

emotional domain, while only 9% were observed to be on track in the learning domain of ECDI. In terms of parental migration, it was found that approximately 9% of the total children had fathers who migrated abroad, while 3.3% had fathers who migrated within the country. The migration of mothers alone accounted for a mere 0.3% of the total children. Additionally, the migration of both parents was reported for only 2% of the total children.

## Association between parental migration and early childhood development and its independent domains

The associations between parental migration and early childhood development and its independent domains are presented in Table 3. Children with fathers who migrated abroad had significantly decreased odds of being developmentally on track in the literacy-numeracy domain (OR: 0.84, 95% CI: 0.73–0.96), physical domain (OR: 0.88, 95% CI: 0.76–0.99), learning domain (OR: 0.80, 95% CI: 0.70–0.91), and overall ECDI (OR: 0.74, 95% CI: 0.54–0.93). Furthermore, when both parents migrated, a significant association was found with decreased odds of the learning domain (OR: 0.86, 95% CI: 0.55–1.35) and overall ECDI (OR: 0.63, 95% CI: 0.48–0.97). Notably, any type of parental migration exhibited significant associations with decreased odds of being developmentally on track in the literacy-numeracy domain (OR: 0.90, 95% CI: 0.80–0.99), learning domain (OR: 0.83, 95% CI: 0.71–0.98), and overall ECDI (OR: 0.85, 95% CI: 0.77–0.95).

Among the adjusted covariates, it was found that poor early ECDI outcomes were associated with several factors (S1–S4 Tables). Specifically, children in Barishal division, those who did not attend early childhood education, and children whose mothers had functional difficulties or belonged to the poorest wealth quintile were found to have lower likelihoods of positive early childhood development. These factors were identified as significant contributors to the observed disparities in ECDI outcomes.

**Table 3. Effects of parental migration on early childhood development in Bangladesh adjusting for covariates.**

| | Literature domain, OR, 95% CI | Physical domain, OR, 95% CI | Socio-emotional domain, OR, 95% CI | Learning domain, OR, 95% CI | Overall Early childhood Development Index, OR, 95% CI |
|---|---|---|---|---|---|
| **Father migration** | | | | | |
| Not migrated | 1.00 | 1.00 | 1.00 | 1.00 | 1.00 |
| Migrated abroad | 0.84 (0.73–0.96)** | 0.88 (0.76–0.99)** | 1.03 (0.92–1.17) | 0.80 (0.70–0.91)** | 0.74 (0.54–0.93)** |
| Migrated within country | 1.03 (0.83–1.27) | 1.00 (0.82–1.23) | 0.97 (0.81–1.17) | 0.97 (0.80–1.18) | 0.97 (0.79–1.17) |
| **Mother migration** | | | | | |
| Not migrated | 1.00 | 1.00 | 1.00 | 1.00 | 1.00 |
| Migrated abroad | 1.43 (0.72–2.82) | 1.26 (0.66–2.41) | 1.30 (0.67–2.52) | 0.69 (0.28–1.69) | 1.51 (0.82–2.75) |
| Migrated within country | 1.16 (0.87–1.54) | 0.83 (0.62–1.10) | 0.94 (0.74–1.20) | 0.90, 0.61–1.31 | 0.80 (0.61–1.05) |
| **Migration of both parents** | | | | | |
| No | 1.00 | 1.00 | 1.00 | 1.00 | 1.00 |
| Yes | 1.15 (0.83–1.59) | 0.78 (0.57–1.09) | 1.04 (0.79–1.37) | 0.86 (0.55–1.35) | 0.63 (0.48–0.97)** |
| **Any type of migration** | | | | | |
| No | 1.00 | 1.00 | 1.00 | 1.00 | 1.00 |
| Migrated one of the parents | 0.90 (0.80–0.99)** | 0.92 (0.82–1.04) | 1.01 (0.91–1.12) | 0.83 (0.71–0.98)** | 0.85 (0.77–0.95)** |

**Notes:** **indicates $p < 0.05$,

*indicates $p < 0.01$

## Discussion

The primary objectives of this study were to examine the impact of parental migration on early childhood development in Bangladesh, specifically focusing on the ECDI and its individual domains. The findings of this study provide compelling evidence of the significant negative effects of parental migration on the developmental outcomes of children in the country. The results indicate that children with fathers who migrated abroad faced notable disadvantages, experiencing a 26% decrease in the likelihood of being developmentally on track overall. This reduction becomes even more pronounced, reaching 37%, when both parents migrate. Furthermore, children whose parents had migrated, irrespective of the type of migration, exhibited a significant 15% decrease in their ECDI scores. These findings underscore the wide-ranging consequences of parental absence due to migration on a child's overall development, emphasizing the critical importance of the early years as these effects can have lifelong implications.

The observed negative correlation between parental migration and poor ECDI scores are comparable to the other study findings in China and other countries [10, 12, 14, 19–21]. This relationship can be attributed to several contributing factors. Firstly, children of migrant parents are at an increased risk of facing challenges related to malnutrition, compromised physical health, and unintentional injuries compared to children whose parents do not migrate [21]. This disparity arises from the lack of parental involvement and support in fulfilling child-care responsibilities, resulting in inadequate supervision and care for the child's well-being [1]. Moreover, the absence of parents due to migration disrupts the crucial parent-child relationship during the formative years, which plays a fundamental role in fostering healthy child development [21]. Parental involvement, support, and nurturing are vital for creating a nurturing environment that promotes cognitive, social, and emotional growth [1, 4, 22]. When

parents are absent for extended periods, children may experience a deficiency in emotional bonding and have limited opportunities for interactive learning experiences, thus hindering their overall development [1, 3, 4].

Furthermore, the emotional strain and stress associated with parental migration can have profound detrimental effects on children's overall well-being [3]. The separation from a parent can trigger feelings of anxiety, sadness, and insecurity, significantly impacting their emotional and cognitive functioning [12, 21]. These emotional disruptions may manifest as challenges in regulating emotions, forming relationships, and adapting to new environments, ultimately compromising socio-emotional development [21]. Another influential factor contributing to the negative association between parental migration and ECDI scores is the economic strain faced by families with migrating parents [9]. The financial pressures and the need for remittances to support the family back home can divert attention and resources away from investing in the child's early development [14]. Limited financial means may result in reduced access to quality early childhood education, healthcare services, and enriching learning materials, all of which are essential for promoting optimal cognitive and physical development [23, 24]. Furthermore, the absence of parents can disrupt the stability of the home environment, leading to alterations in daily routines, changes in caregiver arrangements, and shifts in family dynamics [25, 26]. These disruptions introduce stressors and inconsistencies that negatively impact a child's sense of security and stability, potentially hindering their learning progress and overall development [1, 9].

These findings depict a deeply concerning reality in which parents, driven by unwavering hope to secure a brighter future for their children, unknowingly subject them to adverse consequences. In Bangladesh, there is an increasing trend of parents opting for migration, underscoring the urgent need for comprehensive and targeted interventions. It is crucial for policymakers, educators, and practitioners to prioritize the development and implementation of support systems specifically tailored to address the unique needs of children with migrating parents. Effective initiatives should include the establishment of community-based programs that serve as a lifeline, providing emotional support, counselling services, and educational resources [1]. The government's current initiative to ensure early childhood education, which has been found to have a negative association with poor ECDI outcomes in this study, should be strengthened by addressing the existing challenges [27]. These challenges include improving poor infrastructure and addressing the shortage of skilled teaching professionals [28]. Additionally, it is crucial to prioritize the needs of children from poor households, as they are more likely to have migrating parents and caregivers who may have limited education and knowledge about early childhood care [29]. This study also revealed a correlation between poor ECDI outcomes and children from disadvantaged households, aligning with existing data from LMICs [3, 4]. Furthermore, it is important to focus on children whose parents have disabilities, as this study identified a link between poor ECDI outcomes and such cases, particularly when one parent is disable and the other is migrated [30, 31]. Taking these factors into consideration and implementing targeted interventions will be vital in addressing the specific challenges faced by these vulnerable children and ensuring their optimal early childhood development. These programs should aim to equip these vulnerable children with the necessary tools to navigate the emotional and developmental challenges that accompany parental migration [4]. By ensuring equal access to enriching learning environments, fostering nurturing social interactions, and maintaining consistent routines, children of migrating parents will get the crucial support they desperately require for wholesome development, even in the painful absence of their migrating parents [11].

This study has several strengths, as well as a few limitations. One of the key strengths is that it is the first study in Bangladesh to report on the impact of parental migration status on ECDI. Advanced statistical techniques were employed, and nationally representative survey data were

analyzed, taking into account a range of confounding factors that were selected based on previous literature in low- and middle-income countries. Additionally, the categorization of ECDI followed the recommended guidelines provided by UNICEF. However, a major limitation of this study stems from the analysis of cross-sectional survey data, which means that the reported findings are correlational in nature and not causal. Furthermore, the data recorded in this study relied on self-reported responses with limited options for validating the data. This introduces the possibility of recall bias, although any such bias is likely to be random. Moreover, there are additional factors, such as the quality of early childhood education and dietary patterns, that may impact ECDI and were not considered in the model due to their unavailability in the survey data. Despite these limitations, this study serves to raise awareness among policymakers about the ongoing issue of parental migration and its impact on ECDI. It can inform evidence-based policymaking and the development of programs aimed at reducing the negative effects of parental migration on early childhood development.

## Conclusion

Approximately 29% of children aged 3 to 4 years in Bangladesh were found to be not developmentally on track, as measured by the ECDI. The likelihood of poor ECDI outcomes was 26% higher among children with fathers who migrated abroad. This disparity becomes even more significant, reaching 37%, when both parents migrate. These findings indicate a potential risk where approximately one-third of all children in Bangladesh may face lifelong developmental challenges, with a higher likelihood among children whose parents migrate abroad in search of better opportunities for their families and countries. To address this issue, it is crucial to strengthen early childhood education and social safety nets, ensuring comprehensive early childhood development for all children, with a particular focus on the children of migrant parents.

## Supporting information

**S1 Table. Effects of father migration on early childhood development.**
(DOCX)

**S2 Table. Effects of mother's migration on early childhood development, Bangladesh.**
(DOCX)

**S3 Table. Effects of migrating both parents on early childhood development, Bangladesh.**
(DOCX)

**S4 Table. Effects of migrating one of parents on early childhood development index, Bangladesh.**
(DOCX)

**S5 Table. STROBE Statement—checklist of items that should be included in reports of observational studies.**
(DOCX)

## Acknowledgments

We are thankful to the Jatiya Kabi Kazi Nazrul Islam University where this study was conducted.

## Author Contributions

**Conceptualization:** Shimlin Jahan Khanam, Md Nuruzzaman Khan.

**Formal analysis:** Shimlin Jahan Khanam, Md Nuruzzaman Khan.

**Software:** Shimlin Jahan Khanam.

**Supervision:** Md Nuruzzaman Khan.

**Writing – original draft:** Shimlin Jahan Khanam.

**Writing – review & editing:** Md Nuruzzaman Khan.

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
