## [Decision Letter · Decision Letter 0]

26 Sep 2023

PONE-D-23-17995Effects of Parental Migration on Early Childhood Development of Left-Behind Children in Bangladesh: Evidence from a Nationally Representative SurveyPLOS ONE

Dear Dr. Khanam,

Thank you for submitting your manuscript to PLOS ONE. After careful consideration, we feel that it has merit but does not fully meet PLOS ONE’s publication criteria as it currently stands. Therefore, we invite you to submit a revised version of the manuscript that addresses the points raised during the review process.

ACADEMIC EDITOR: Please insert comments here and delete this placeholder text when finished. Be sure to:Situate/relate this study in relation to earlier similar studies, especially in the discussion.Ensure that the reference style of the journal is adhered to all through the manuscript.==============================

We look forward to receiving your revised manuscript.

Kind regards,

Folusho Mubowale Balogun

Academic Editor

PLOS ONE

Reviewers' comments:

Reviewer's Responses to Questions

**Comments to the Author**

1. Is the manuscript technically sound, and do the data support the conclusions?

Reviewer #1: Yes

Reviewer #2: Yes

2. Has the statistical analysis been performed appropriately and rigorously? 

Reviewer #1: Yes

Reviewer #2: Yes

3. Have the authors made all data underlying the findings in their manuscript fully available?

Reviewer #1: No

Reviewer #2: Yes

4. Is the manuscript presented in an intelligible fashion and written in standard English?

Reviewer #1: Yes

Reviewer #2: Yes

5. Review Comments to the Author

Reviewer #1: This study examine the influence of parental migration on child welfare and early development outcomes in Bangladesh. The study uses a large sample which is one of the strengths of this paper. The findings especially around the impact of both parents migration on child development is worthy of dissemination to a wider audiences and specialist researchers, policy practitioners, CSOs engaged in child welfare and development. The findings have relevance to other low and middle income countries.

I only have minor comments as I feel this study is clearly written the approach has been adequately described and can be easily followed by readers.

a)I would advise the authors to closely follow the citation style of the journal especially citation in text.

INTRODUCTION

b) The authors zoom a little too quickly to Bangladesh even when a 'funnel approach' would help readers set the context for this paper. The 'funnel approach' is setting the global scene, a regional perspective and finally anchoring the study in Bangladesh.

METHODS

c) Was this a secondary data analysis or did you use primary data. This is not perfectly clear and is a little ambiguous. I would be clearer in bringing this out.

d) Have similar data sets been utilized in other countries? It would help to reference them here if this is the case.

RESULTS

I would use 'demographic' characteristics instead of 'background' statistics.

DISCUSSION

I would do more to link this study with previous efforts. The reference list alone betrays this deficiency.

Overall, this appears to have been a competently conducted study and would recommend publication after minor revisions

Reviewer #2: General comments

Thank you for the opportunity to review this manuscript which focuses on the effect of parental migration on early childhood development in Bangladesh. The importance of studying this subject at individual country level is aptly stated by the authors.

The manuscript is generally well written bringing out the objectives, results, and conclusion of the research concisely.

The subject matter is a topical issue which will be of interest to both clinical and non-clinical specialties in different countries.

Minor issues

1. The authors should revise the language and syntax to improve readability e.g., “on-track status” could be inserted after “were indicators” in line 4, page 12. The discussion section also contains some the language and syntax issues for e.g., line 4, page 22.

2. “Bangladesh” does not need to reflect in the title of Table 1.

3. The results in Table 2 suggest that the first row ought to be captioned as “Children who are developmentally on track for indicated domains”. The authors may however clarify if this is not the intended message to readers.

4. Reference 29 should be better cited for reproducibility.

Major issues

1. The authors should clarify how the sample size of 8,833 was obtained out of the total 23,099 eligible children to avoid confusion.

2. The authors stated that “We analyzed a total of 8,833 children, with 4,513 children aged 3 years (24-36 months, N=4,513) and 4,320 children aged 4 years (37-48 months, N=4,320).” For clarity, the authors should re-write this age bracket to avoid the confusion that may result from interpreting 24 months as 3 years or 37 months as 4 years.

3. The ages of the children studied should be consistently stated in the manuscript for e.g., the Conclusion section reports that children 3 – 4 years were studied in contrast to 2 – 4 years reported in the Methodology section.

4. The result on the association between parental migration and ECD is that children with fathers who migrated abroad had significantly decreased odds of being developmentally on track in the overall ECDI (OR: 0.74, 95% CI: 0.54-0.93). It is however stated differently in the abstract section as a decrease in the odds of a poor ECD. This difference should be reconciled.

Overall, the manuscript is presented in a clear and logical manner that keeps the reader interested.

6. PLOS authors have the option to publish the peer review history of their article (what does this mean?). If published, this will include your full peer review and any attached files.

Reviewer #1: No

Reviewer #2: No

---

## [Author Response · Author response to Decision Letter 0]

19 Oct 2023

We have added a response to reviewers file where we provided a point by point response to all comments we received from the reviewers and the editor.

---

## [Editor Report · Decision Letter 1]

6 Nov 2023

PONE-D-23-17995R1Effects of Parental Migration on Early Childhood Development of Left-Behind Children in Bangladesh: Evidence from a Nationally Representative SurveyPLOS ONE

Dear Dr. Khanam,

Thank you for submitting your manuscript to PLOS ONE. After careful consideration, we feel that it has merit but does not fully meet PLOS ONE’s publication criteria as it currently stands. Therefore, we invite you to submit a revised version of the manuscript that addresses the points raised during the review process.

ACADEMIC EDITOR:Most of the concerns and comments have been addressed.

The outstanding corrections are:

- Background: The beginning of the 3rd and 4th sentences in 2nd paragraph should be rephrased.

The word 'however' at the beginning of both sentences is distorting the flow of  the paragraph.

- Results: There is disparity in the percentage of children from the rural area in the text and table 1.==============================

We look forward to receiving your revised manuscript.

Kind regards,

Folusho Mubowale Balogun

Academic Editor

PLOS ONE

Journal Requirements:

Additional Editor Comments:

Most of the concerns and comments have been addressed.

The outstanding corrections are:

- Background: The beginning of the 3rd and 4th sentences in 2nd paragraph should be rephrased.

The word 'however' at the beginning of both sentences is distorting the flow of the paragraph.

- Results: There is disparity in the percentage of children from the rural area in the text and table 1.

---

## [Author Response · Author response to Decision Letter 1]

8 Nov 2023

We have uploaded a MS word file where we provided point-by-point response to editor's comments.

---

## [Editor Report · Decision Letter 2]

13 Nov 2023

Effects of Parental Migration on Early Childhood Development of Left-Behind Children in Bangladesh: Evidence from a Nationally Representative Survey

PONE-D-23-17995R2

Dear Dr. Shimlin Jahan Khanam,

We’re pleased to inform you that your manuscript has been judged scientifically suitable for publication and will be formally accepted for publication once it meets all outstanding technical requirements.

Kind regards,

Folusho Mubowale Balogun

Academic Editor

PLOS ONE
---

## [Editor Report · Acceptance letter]

20 Nov 2023

PONE-D-23-17995R2 

Effects of Parental Migration on Early Childhood Development of Left-Behind Children in Bangladesh: Evidence from a Nationally Representative Survey 

Dear Dr. Khanam:

I'm pleased to inform you that your manuscript has been deemed suitable for publication in PLOS ONE. Congratulations! Your manuscript is now with our production department. 

Kind regards, 

on behalf of

Dr. Folusho Mubowale Balogun 

Academic Editor

PLOS ONE